# Approaches for Management and Valorization of Non-Homogeneous, Non-Recyclable Plastic Waste

**DOI:** 10.3390/ijerph191610088

**Published:** 2022-08-15

**Authors:** Stefano Gazzotti, Beatrice De Felice, Marco Aldo Ortenzi, Marco Parolini

**Affiliations:** 1Department of Environmental Science and Policy, University of Milan, Via Celoria 26, 20133 Milan, Italy; 2Department of Chemistry, University of Milan, Via Golgi 19, 20133 Milan, Italy

**Keywords:** plastic waste, recycling, non-recyclable plastics

## Abstract

The environmental accumulation of plastic wastes has become one of the most discussed topics in the scientific community. The development of new strategies to tackle this issue is of crucial importance, and different approaches are being investigated to effectively reduce plastic waste generated by improper or inefficient disposal. In addition to the efforts addressing the development of biodegradable plastics, the research is currently focused on the development of innovative recycling approaches. Indeed, although most plastic materials are potentially recyclable, only 15% of the worldwide plastic waste is currently recycled, while the remaining 85% is usually incinerated to recover thermal energy or landfilled. The hurdles to efficient recycling come from improper management of end-of-life plastic goods. Moreover, the highly heterogeneous nature and versatility of plastic and polymeric materials have led to the development of multilayered materials, composites, blends and many other different species, whose management and/or reprocessing to yield high-value products is extremely challenging. Thus, although these materials are extremely valuable from an industrial point of view, they add a high degree of complexity to the recycling process because each one of them is different from the other, but they cannot be separated efficiently. The aim of the present review is to return a comprehensive overview of environmental and management issues related to the complex and heterogeneous mixture of plastic waste that is generated at the end of the sorting procedures in Italian plastic recycling plants, the so-called ‘Plasmix’. This review lists the difficulties and limitations related to the management of non-recyclable Plasmix and highlights the strategies for the proper, sustainable and valuable use of this plastic waste.

## 1. Introduction

Over the past 50 years, the role and importance of plastics in our economy have consistently grown because they incalculably contribute to tackle most of the challenges that our society has to face. The versatility, biological neutrality, mechanical strength, ease of industrial production and low production costs have made plastics among the most extensively used materials worldwide. Whilst in 1950, the plastic demand stood at 2 Mt [1], in 2020, it increased up to 367 Mt [2], and it is expected to quadruple by 2050 [3]. In accordance with the increasing plastic demand, it has been estimated that, up to 2017, 8300 Mt of plastics has been produced since the 1950s [4]. Plastics entered mainstream demand and use for a wide range of products, most of which have short to average lifespans, including packaging, clothing, automotive, electrical and electronics fields [2]. Despite the countless socio-economic benefits, the continuative production, use and disposal of plastics contribute to the so-called ‘plastic waste problem’ that is characterizing the human–environment relationship. It has been estimated that, since 1950, an amount of 6300 Mt of plastic waste has been generated. Up to date, most of such wastes have been disposed of in landfill; whether the current practices of disposal continue, it has been estimated that, by 2050, 12,000 Mt of plastic waste will enter landfills or the environment [1]. The last scenario is considered particularly worrisome because, once in the environment, plastic wastes can pose a risk to wildlife and ecosystems. Several studies have demonstrated the presence of a wide array of large-size plastic wastes, with different shapes, sizes, colors and polymeric compositions, in both aquatic and terrestrial ecosystems worldwide [5,6]. The presence of these wastes is of particular concern because of their aesthetic and biological impact, but also for the consequence of their breakage, fragmentation and degradation due to chemical, physical and biological processes that lead to the formation of the so-called micro(nano)plastics. Micro(nano)plastics include nanoplastics (NPs), i.e., any plastic items in the 1 to <1000 nm size range, and microplastics (MPs), i.e., any plastic item in the 1 to <1000 µm size range [7], which emerged as a ‘novel’ topic in environmental studies because of their widespread distribution and potential toxicity [8].

To reduce the loss of valuable material and the impact of plastic waste, the development of advanced recycling techniques and the use of biodegradable plastics can be considered valid alternatives. As the latter approach still constitutes a market niche, the most challenging topic remains the management of plastic waste. PlasticsEurope has reported that about 29.5 Mt of plastic waste has been collected for recycling in Europe in 2020, almost 10% higher than in 2017 [2]. Only 34.6% of such waste is recovered for recycling, while the remaining part is intended for landfilling (23.4%) and energy recovery (42.0%). However, a strong reduction in plastic disposal in landfill has been observed from 2006 to 2020 (−46.5%), coupled with an increase in energy recovery (+77%) and recycling processes (+117%). Despite these promising trends, it has been estimated that 95% of the value of plastic packaging (ca. EUR 70–105 bn per year) is lost from the economic system after the first-use cycle [9]. Figure 1 reports a schematic representation of the plastic materials’ life cycle.

Thus, to prevent economic consequences, the growing plastic waste generation and improper disposal must be tackled. The European Directive on packaging and packaging waste (CE/62/94) has imposed a recycling target that currently requires 22.5% of waste plastic packaging to be recycled. This goal is proposed to increase to 55% by 2025, with the final target that all plastic packaging on the EU market will be recyclable by 2030 (EC, 2018). Thus, there is an urgent need to improve the collection, sorting and recycling efficiency of plastic waste in order to create valuable markets for recycled and renewable plastics. Today, the study and development of plastic waste recycling is a hot topic, covering many different scientific fields. For instance, a recent study showed the effect of new product design and source-separation systems on the efficiency of recycling to obtain a potentially closed loop of food-grade quality packaging [10]. Moreover, a second study proposed a tribo-electrostatic separation to increase the efficiency in separating Polystyrene (PS), Polyvinylchloride (PVC), Acrylonitrile Butadiene Styrene copolymer (ABS) and Polypropylene (PP) [11]. In contrast, studies concerning technical solutions for recycling complex mixtures of plastic waste are still limited, although [12] indicated some possible trends and opportunities about mechanical properties of Polyethylene (PE)-PP waste mixed with sawdust. Moreover, ref. [13] studied a PE-PP mixture representative of an average of waste polyolefin plastics found in recycling facilities, mixed with wood using different kinds of compatibilizers, such as Fusabond and Orevac families (roughly 55% wood, 40% PE-PP and 5% compatibilizer), obtaining some promising results in terms of mechanical properties of the composites. Further, other blends coming from waste polymers were investigated, such as waste Polyethylene Terephthalate (PET)-PP blends [14], but no complex mixture has been tested. Despite these findings, increasing the recycling rate is becoming more difficult because waste streams from plastic packaging recovery facilities are very heterogeneous and difficult to separate [15]. To date, the sorting of plastic wastes allows for the separation and regeneration of PET, PE and PP [16]. However, a notable amount of non-homogeneous fractions of mixed polymers arising from mechanical treatment remains: in Italy, such fraction is called ‘Plasmix’ [16].

Plasmix includes the rejected fraction of sorting, and it consists of a complex mixture of different materials that varies over time and depends on the separation efficiency [15]. Basically, Plasmix is the mixture of materials that remains after the sorting of the most easily separable and recyclable plastics, consisting of PET and the most valuable polyolefin fractions, mainly high-density PE (HDPE) and PP. The composition of Plasmix is closely associated with the type of differentiated waste collection management. Therefore, it varies not only from state to state but even from municipality to municipality. Its composition is, therefore, not well defined but is mainly composed of PP, HDPE and low-density PE (LDPE). Plasmix consists of two different fractions coming from plastic waste mechanical treatments, namely the under-sieve from the size separation equipment and the final residues from the whole mechanical sorting operations. Overall, in the Italian context, Plasmix is composed of a mixture of different materials, including plastic (57%), paper and cardboard (10%), wood (3%), textiles (3%), inert and others (27%) [17]. Among plastics, 60–70% is represented by polyolefin mixture (typically PE is more than PP), 10–15% of water derived from washing processes or from rain, 4–5% PET derived from food boxes or from opaque white bottles, 2–4% PS and by low amounts of Polyamides (PA), ABS, Expanded Polystyrene (XPS) and Polyurethanes (PU) or other polymers. The further separation of the different polymeric fractions constituting the Plasmix is generally difficult and economically not advantageous. These heterogeneous materials cannot find any application other than being used as secondary solid fuel in incinerators or cement factories, exiting from the circular value chain of recycling. In Italy, Plasmix is mainly incinerated (57%), used as a substitute for coal burning in cement kilns (27%) or landfilled (16%) [16]. Considering the increasing complexity of plastic packaging and the high efficiency of the recycling process, the amount of plastics that are separated and cannot experience a valuable regeneration has constantly increased, growing from 10–15% (1998–2000) to ≈50% today. In 2017, Lombardy collected 192,207 t of plastic waste (+9.3% compared to 2016), corresponding to 19.2 kg pro capite, of which about half is composed of Plasmix. Despite the high quantities of Plasmix produced every year, limited scientific literature is available concerning its possible modifications and valorization, and just a few possible large-scale industrial applications are currently under study. For instance, in March 2019, COREPLA and ENI signed an agreement aimed at developing the production of hydrogen from Plasmix, while in November 2018, an Italian patent company, VGM Patent, launched a crowdfunding campaign to create adhesives for wood shavings from Plasmix. Some trials have also been recently made regarding the production of 3D printing filaments by Revet Recycling and R3direct and regarding the use of polyolefin mixtures as additives for asphalts by Tesco in England as a demonstration that the fate and valorization of Plasmix have become a pressing issue. Lastly, a recent study aimed at evaluating the effects of the catalysts on the polymers’ degradation temperatures and to determine the main compounds produced during pyrolysis of a synthetic mixture of real waste packaging plastics representative of the residue from a material recovery facility [15].

This review lists the different techniques currently available for sorting and recycling plastic waste and first focuses on the difficulties and limitations related to the management of non-homogeneous, non-recyclable plastics (i.e., Plasmix), highlighting the strategies for the proper, sustainable and valuable use of this plastic waste.

## 2. Sorting of Plastic Waste

The problem of plastic recycling strongly depends on the great amount of plastic types. The collection of waste does not end up in the separation of the different plastic types. Therefore, compatibility issues have to be resolved before the recycling process can start. In a typical collection, there are a lot of different plastic families and grades within the same family, so their segregation has to be performed to sort out various materials. Moreover, the introduction of one polymer into another, for example, in the production of multilayer films, often leads to a reduction in the properties of the recycled material. The reason behind this observation comes from the fact that each polymeric material has its own specific properties and processing features that may not be compatible with the ones of other materials. For example, thermal properties can influence the compatibility between different polymers [18]: from one side, each material is characterized by specific thermal transition temperatures that define the thermal profiles to be employed for the processing. Notable differences existing between different materials can significantly hurdle reliable and efficient processing, resulting in heterogeneous mixtures. In addition, different polymers often display remarkably different degradation temperatures: this means that the temperature required for proper processing of a certain material is too high for another one that will eventually degrade and yield an unusable recycled material. On the other hand, if such a temperature is too low to allow the melting of some polymeric fractions, the latter will be present as an unmelted solid particle in the molten matrix, leading to defects and loss of mechanical properties. A remarkable example of this concept is represented by the contamination of polylactic acid (PLA) in the PET recycling stream. Even very small amounts of PLA (<0.1% concentration) can result in an unusable recycled PET (rPET) resin [19].

Another remarkable issue when dealing with the combination of different polymeric materials is represented by their relative miscibility. The wide applicability of polymers in many areas comes, among other factors, from the differences in their chemical nature. Polyamides, polyurethanes, polyesters, polyolefins and so on are very distinct families of polymers, and each one of them is characterized by very specific chemical bonds to link the repeating units constituting the polymeric chains. In addition, each family has various subclasses (aliphatic, aromatic, semi-aromatic and so forth), generating virtually infinite combination possibilities and, consequently, a very wide span of properties that make polymers such a versatile group of materials. Even if this versatility is a strong selling point for polymeric materials, it can generate problems during the recycling processes because the great chemical difference between two or more species could result in scarce miscibility [20]. Indeed, the lack of favorable secondary interactions between two or more polymers makes them immiscible and ends up in the formation of a heterogeneous material with extended phase separation when the two components are eventually mixed [21], also leading to very poor miscibility in polymers that should be apparently well miscible due to their similar chemical nature, such as PE and PP [22]. A relevant example of this concept is represented by PET-PE mixtures. While the two pristine polymers are recyclable, their mixing is not efficient, and the resulting material is brittle due to extensive phase separation [23].

For all the aforementioned reasons, it becomes necessary to sort and separate the various kinds of plastic materials in order to improve the recycling process and make it applicable and reliable. As this step is so important, a lot of different methodologies have been developed based on different analytical techniques.

### 2.1. Laser-Induced Break Down Spectroscopy

Laser-induced breakdown spectroscopy (LIBS) is an analytical technique based on pulsed laser sources [24]. It can be applied for the identification of different kinds of plastic materials. The LIBS technique is based upon the analysis of the atomic emission lines generated by focusing high-energy laser radiations on the sample surface [25]. A laser-produced plasma emission is recorded for spectral analysis of various kinds of plastics to yield an emission spectrum that can be used as a fingerprint for the identification of the various materials. Mainly six kinds of plastic materials, i.e., LDPE, HDPE, PP, PS, PET and PVC, can be identified by this technique. Optimization of the analysis conditions is carried out on pristine materials and then applied to identify the components of the waste mixture.

### 2.2. Tribo-Electric Separation

Electrostatic separation is a broadly applied technology for the sorting of particle blends with particle sizes of around 5 mm [26]. At the industrial level, the tribo-electric separation particles made of insulating materials (i.e., plastics, among others) using a rotary tube is an efficient technology employed in waste recovery and mineral industries. A separation device based on tribo-electric technology allows materials to be sorted on the basis of a surface charge transfer phenomenon [27]. For example, the so-called tribo-cyclone is a device that applies a centrifugal force to the particles that eventually charge due to their acceleration and friction against the charging surface [28]. This charging surface is a plastic lining made from a material that has to be carefully chosen depending on the materials that have to be separated. Cu electrodes are placed at the end of the system in order to separate the particles depending on the charge. This process is widely used for the sorting and purification of granular materials resulting from industrial plastic wastes [29]. Similar techniques can also be applied to separate plastics from metallic elements and other non-recyclable components. This type of separation technique can be used to separate nearly all plastic materials. The main limitation is that it can be exploited for the separation of materials with a maximum particle size of 2–4 mm.

### 2.3. FTIR (Fourier Transformed Infrared)

The FTIR technique can be used for the identification of different types of polymers and plastic materials. The analysis works through the irradiation of the sample by IR radiation and by recording the spectrum generated by the motion of the functional groups in the molecule excited by the radiation. The FTIR analysis is very versatile as it can be applied to liquid, gaseous and solid specimens. An FTIR spectrometer is able to collect high spectral resolution data over a wide spectral range. This confers a significant advantage over a dispersive spectrometer, which measures intensity over a narrow range of wavelengths at a time. In addition, the analysis is highly reliable, with extremely good sensitivity. For this reason, FTIR spectroscopy can also be used to examine the structural variations [30]. This technology is, therefore, also applicable to analyzing the structural change during the recycling of polymers. The standard IR detector can be replaced or complemented by hyperspectral imaging spectroscopy (HIS) to recognize a full-shape product or by an X-ray fluorescence detector to recognize heavy elements such as Cl and Br. HIS is a fast and non-destructive technique that can also be utilized for the analysis of solid particulate systems in terms of composition and spatial distribution, also in different fields, such as in food and pharmaceuticals [31]. These advances are reported to allow challenging sorting, e.g., HDPE/LDPE, PET/PLA, or black products that cannot be identified with conventional NIR detectors [32].

### 2.4. Froth Flotation Method

In 1978, Alter [33] first described the recovery of plastics by froth flotation, depending on their critical surface tension. Even if many other separation techniques are available, froth flotation is one of the simplest and cheapest methods that has been successfully applied in several different fields [34]. In the case of plastics, froth flotation can be applied to separate post-consumer PET (Polyethylene Terephthalate) from other packaging plastics with similar density [18]. Because of the generally hydrophobic nature of all plastic/polymer material, froth flotation can be challenging as air bubbles’ presence in the material makes the material float. Separation of most plastics is, however, possible, given some exceptions: for example, LDPE and HDPE are not separable by this technique [35]. In froth flotation, wetting and frothing agents are necessary for the recovery of plastics. Calcium lignin sultanate as a wetting agent and pine oil and MIBC (methyl isobutyl carbinol) as a frothing agent are some of the most significant examples of these species. For example, pine oil can be exploited as a frothing agent for the recovery of PVC, while MIBC for PET recovery.

### 2.5. Magnetic Density Separation

Magnetic density separation is a technology able to sense even small changes in the physical properties of materials and, therefore, can be used for the production of high-purity material from complex streams of post-consumer waste. This technology can be helpful in the separation of useful plastic from waste with minimized residue material [27]. MDS is a physical separation method based on the differences in density of the materials [36]. This technique can be applied for the separation of the various types of PP, LDPE and HDPE from each other and from contaminating materials such as wood, rubbers and minor amounts of metals. MDS is potentially very cheap because it allows the separation of a complex mixture into many different materials in a single step. The entire process is performed as the mixture flows through a channel, and separation occurs in seconds into different layers. The MDS setup consists of four different steps: (i) Wetting, (ii) Feeding, (iii) Separating and (iv) Collecting. The materials are first wetted with boiling water for a minute. Then, the wetted particles are fed into a stainless-steel box with openings of 1 mm. When the lid of the box is open, the particles start flowing into the separation channel with the mainstream; here the density of the material plays a major role. The speed of the flow in the separation channel has to be optimized in order to allow for the optimal separation of the components of the mixture. At the end of the separator, the different materials are collected. More progress in accurate sorting is to be expected soon. Nowadays, sorting technologies are able to produce only a limited fraction of monostreams while leaving a lot as mixed plastic waste. On a general level, it is estimated that between 60% and 70% of the material can be sorted as monostreams [37]. Monostreams can be sold for reprocessing and blending into new plastic products. It should be noted, however, that this reprocessing often starts with a finer sorting that further reduces the volume of plastic that is effectively recycled.

## 3. Recycling of Plastics

There are four main approaches for the recycling of plastic wastes, namely primary, secondary, tertiary and quaternary recycling (Figure 2).

Sorted plastics are usually not suitable for direct reprocessing because, for example, they may require cleaning to remove dirt and other contaminants, e.g., from packaged food or from mixed consumer waste. The cleaning is generally imperative for secondary (mechanical) recycling but might be important for tertiary (chemical) recycling as well. Plastic waste is generally cleaned with hot or cold water, with the assistance of caustic agents or detergents [38]. The cleaning is often integrated into the sorting chain, e.g., after shredding and combined with a sink-float sorting step. Such washing can be costly as it requires dedicated washing equipment but also a drying step and a wastewater treatment. It may, furthermore, lack the efficiency required. For instance, odorous components appear to be only partially removed by a caustic wash; the most apolar components require the use of detergent or organic solvent to be removed [39,40]. Various dry-cleaning approaches are also being investigated in an attempt to avoid the cost and water demand of conventional wet cleaning. They vary from mechanical cleaning with compressed air assisted with mechanical action such as a rotor disk, scrapping or fluidized sand bed [41]. These dry-cleaning techniques are reported to match the effectiveness of a caustic wash. The recycling technique of choice does play an important role in the generation of a new polymer as it deeply affects the quality of the final material. Every technique has its own advantages and disadvantages. When the material undergoes a recycling process, it often starts losing some of its properties in terms of tensile strength, wear properties and dimensional accuracy, which are usually connected to the reduction of molecular weight caused by the high temperatures required for the recycling.

### 3.1. Primary Recycling

Primary recycling, better known as re-extrusion or closed loop process, is the recycling of an uncontaminated, single type of polymer having properties similar to virgin material [41]. This process is based on the use of scrap plastics that have similar features to the original products. Materials suitable for primary recycling are usually not the ones coming from post-consumer waste as they have to be clean or semi-clean with the lowest amount of contaminants possible. Generally, primary recycling involves injection molding and other mechanical recycling techniques.

Primary recycling has the undisputed advantage of ending up in the manufacture of goods with the same quality and performance as the ones coming from virgin material. The condition for primary recycling to be efficiently carried out is that the recycled material has to be extremely clean and comparable in quality to virgin material. Unfortunately, this is usually not the case for post-consumer waste and, therefore, the waste volumes suitable for primary recycling are very limited.

### 3.2. Secondary Recycling

Secondary recycling involves the mechanical transformation of plastic waste into usually lower-quality materials. Secondary recycling involves different steps such as cutting/shredding, contaminants separation and flakes separation by floating. After these steps, the single polymer plastic material is processed and milled together to form chips. Thus far, plastic recycling consists mainly of mechanical recycling and is focused on the three dominant packaging polymers PE, PP and PET. The recycling processes rely on cautious sorting of the clean and pure monostream fraction, compounding it to granules, and, occasionally, blending them with a virgin polymer of the same family, together with compatibilizers and additives to mitigate the shortcomings of the recycled material [42,43]. Even when secondary recycling is very efficient and conducted on very clean and purified waste, it is limited to a few recycling cycles. For example, PET is generally recycled/downcycled once, from bottle to textile: to increase the number of possible recycling cycles, additives must be used to limit its downgrading. PP should be able to sustain more cycles as it is chemically more stable than PET. However, it is practically recycled/downcycled once to textile and playground equipment due to the presence of impurities in the PP recycling stream. Generally, mechanical recycling is associated with downcycling of the material as the material itself undergoes strong chemical-physical stresses that undermine its integrity, resulting in recycled materials with generally lower performances than virgin ones. This is particularly true for post-consumer plastics because the sorted material may not be as pure as it would be necessary as it would be collected as market-average grade and, therefore, may not meet the requirements for high-end applications. In addition, plastic products often contain additives such as fillers, antioxidants, plasticizers, pigments, flame retardants and so forth, of different chemical compositions and amounts depending on the final applications. Recycled materials will, therefore, contain the market average of these additives. Moreover, sorted waste may not have the same purity as virgin material: the sorting process is often imperfect and results in the presence of varying amounts of foreign materials that could eventually affect the recycling process. As polymers of different chemical natures are practically immiscible with one another, these polymer impurities tend to segregate into small foreign domains that create weak spots in the recycled material. In order to reduce this problem, compatibilizers can be added to polymer mixtures in order to improve the miscibility between the various components. These compatibilizers can carry two or more short chain segments with a structure based on the two or more mixed polymers [38]. Alternatively, the compatibilizers can contain a main chain that resembles the target matrix and a reactive end group that can react with the functional group of the polymer impurity, for instance, one that can react with an alcohol group of PET or EVOH (ethylene-vinyl alcohol) polymers. In addition, recycled polymer chains may be partially degraded, e.g., through oxidation or UV radiation upon use or through thermal degradation upon repeated hot processing. Indeed, polymeric materials are usually processed at high temperatures going from 160 °C for hydrocarbons to 300 °C for PET. Such processing temperatures can be difficult to sustain, particularly when the polymer chains have already been partially degraded during use. As a result, the degraded polymer chain may exhibit minor amounts of polar groups that need compatibilization arising from either depolymerization or the oxidation or both. The reaction of these groups can end up in recycled polymers exhibiting an increased Mw and higher viscosity. Lastly, some applications such as food packaging are not allowed to use materials that could be contaminated by traces of toxic impurities. Mechanical recycling to make plastics for food packaging is then a challenging option. These shortcomings are particularly relevant for post-consumer wastes. It has to be mentioned that, even if these problems are usually connected to fossil sources-derived plastics, they are also present in bio-based materials [44]. It also has to be considered that the addition of compatibilizers and other additives ends up in materials with lower purity that eventually are even more difficult to reprocess and recycle again. One solution can be found in solvent extraction recycling methodologies. The generic framework of plastic recycling by solvent extraction includes the removal of impurities, dissolution (homogeneous or heterogeneous dissolution), and re-precipitation or devolatilization. Specifically, the polymer(s) is dissolved in the solvent(s), and then each polymer is selectively re-precipitated. Ideally, when a solvent can dissolve either the target polymer or all the other polymers except the target one, it can be used for selective dissolution [45]. This strategy is still under development but looks promising for future applications at an industrial level.

The advantage of secondary recycling comes from that it is a cheap technique as it mainly relies on mechanical processing, which is generally easily accessible. In addition, secondary recycling could provide high-quality recycled materials with mechanical and chemical features on par with the ones coming from primary recycling. One of the most relevant drawbacks of secondary recycling comes from the fact that it is highly sensitive to the quality of the sorted post-consumer materials. If the plastic waste is not pure or clean enough, then its recycling is inefficient or even impossible. Another drawback regards the stability of the polymer, which affects the number of cycles a material could sustain. While glass and metals could virtually be recycled an unlimited number of times, polymers usually degrade into lower molecular weight products that are downgraded to less demanding applications.

### 3.3. Tertiary Recycling

As discussed above, mechanical recycling presents a lot of drawbacks and can have limited applicability both in terms of recyclable fraction and final application of the recycled material. Chemical (tertiary) recycling may help extend the market reach and/or the recyclable fraction. Chemical recycling is based on the fact that some polymers can be depolymerized back to their monomers, and others can only be converted to a general feedstock [46,47]. The general guidelines to understand if a polymer is suitable for chemical recycling can be related to the relative heat of depolymerization. Polymers showing a low-to-modest heat of depolymerization (ΔH < 70 kJ/mol of broken bonds) generally consist of condensation polymers, while the ones showing a high heat of depolymerization (ΔH > 70 kJ/mol) are usually addition polymers. This distinction indicates that polyolefins are suitable for cracking back to general feedstocks, while polyesters and polyamides are perfect candidates to be depolymerized back to their monomers. Condensation polymers are formed by nucleophilic substitution reactions to yield ester (–C(O)O–), amide (–C(O)-NH–) or urethane/carbamate bonds (–C(O)(NH)O–). Most of them are prone to opening through hydrolysis, transesterification or transamidation. For example, PET is commonly depolymerized by alcoholysis, i.e., methanolysis (with methanol) to dimethyl terephthalate ester [48]. The reaction is generally carried out at an elevated temperature (~200 °C) in the presence of catalysts, traditionally a Lewis metal salt such as Zn acetate [49]. The methanolysis reaction is often difficult to optimize, especially regarding the purification of the products. For this reason, it is common to target low-viscosity oligomers that can be fed back to the polymerization reactor. Polyamides can be depolymerized at high temperatures in the presence of acidic or basic catalysis. This process is usually limited to the depolymerization of Nylon 6 to caprolactam, while it is scarcely applied for the depolymerization of Nylon 66 back to hexamethylene diamine and adipic acid. Pyrolysis can also be applied for the depolymerization of Nylon 6 to caprolactam [50]. Polyurethanes can also be depolymerized through hydrolysis, alcoholysis, glycolysis and aminolysis [51]. Their depolymerization does not yield the starting monomers (i.e., diisocyanates) but rather oligomers that can be reused for the synthesis of new polyurethanes. It has to be mentioned that all these depolymerization strategies can be challenging both from an experimental (i.e., reaction and purification of the products) and economic point of view. Polyolefins cannot be depolymerized back to their monomers. Their tertiary recycling requires harsh pyrolysis conditions ending up in the formation of paraffinic/olefinic waxes at low temperatures (around 450 °C) or olefin-rich gas at higher temperatures (around 700 °C) [52,53]. These pyrolysis products can be used either as fuels or as chemical feedstock and cracked into lower olefins.

The main advantage of chemical recycling comes from the possibility of turning a material into valuable feedstock. This approach is particularly significant when dealing with materials that have lost most of their mechanical properties and chemical integrity and, therefore, cannot be employed again for the manufacture of goods. One of the most relevant drawbacks of chemical recycling regards the current applicability of this approach. While it can be conveniently applied to easily depolymerizable polymers (for example, polyesters or polyamides), it sees limited applicability for other polymers such polyolefins that are characterized by higher chemical resistance. Another drawback of tertiary recycling comes from the reaction yields: even if the process is highly efficient, it is almost impossible to convert 100% of the starting material into feedstock. Further, tertiary recycling is affected by the purity of the starting material, and it could, in principle, turn the process into a non-convenient approach from an economical point of view. In addition, chemical recycling usually requires solvents and chemicals that could represent a threat to the environment.

### 3.4. Quaternary Recycling

Unsorted plastics still mixed with unsorted textile, paper/cardboard and other organic fractions can be processed and upgraded to hydrocarbon fuel by means of gasification to synthesis gas (or syngas), a mixture of H_2_ and CO followed by syngas conditioning and conversion to fuel or chemicals. Gasification technologies can, of course, be applied to mixed plastic waste or even well-sorted plastic waste. However, gasification technologies are expensive and, thereby, require large-scale applications. Furthermore, they deliver a low-value product, syngas, which needs further conditioning and conversion to get to hydrocarbons. Together with energy recovery, another way to destroy plastic waste is biodegradation. This process allows polymers to be turned into low molecular weight species such as CO_2_, H_2_O or biomass and is possible only for some types of polymers. The biodegradation ends up in the waste of energy embedded into the chemical bonds of the polymers, but, in principle, it could allow the reduction of the accumulation problems related to plastic materials in the environment.

Quaternary recycling can be considered the last resort for the recycling of plastic materials and can be applied in all these situations in which the other three approaches would not be applicable. As already highlighted, quaternary recycling, therefore, allows the accumulation of plastic waste in the environment to be reduced, resulting, at the same time, in the production of energy. The disadvantages of quaternary recycling involve, first of all, the loss of the chemical energy stored in the polymer bonds. In addition, environmental concerns can arise, given the release of greenhouse gases and other byproducts.

## 4. Complex Mixtures of Plastic Wastes

As discussed above, the best strategy for the recovery of plastic waste would be, in principle, secondary recycling. However, the possibility of efficient and reliable secondary recycling is strongly connected to the quality of the waste itself. In order to guarantee a high-quality recycled material, the sorted waste has to be pure in its chemical composition and clean from impurities. Therefore, the waste quality is a mandatory aspect for the successful outcome of the recycling process. As previously stated, post-consumer plastic is intrinsically heterogeneous and, thereby, of undefined quality. It consists mainly of PE, PP and PET, and a great variety of plastic items besides containing variable amounts of foreign materials (e.g., foreign polymers, additives and other contaminants). For instance, post-consumer plastic bottles, trays and films have been shown to consist of 75 to 90 wt % dominant polymer (PE, PP, PET or PS), 5–15 wt % foreign polymers and paper and 5–15 wt % residue. The foreign material and residue were mainly encountered in the cap/lid and labels [54]. The material heterogeneity is even larger for multilayered films, as the main polymer was found to account for only 55 wt % of the film. In addition, the different additives present in the polymers can play a relevant role in making post-consumer waste even more complex. Thus, extraction methodologies have been developed for the removal of these additives from polymeric materials [55]. This same process can be employed for the actual recycling of the different polymeric species in the mixture, and it is known as recycling by solvent extraction [45]. Overall, the recycling problem is usually connected to the composition of the discarded material, which mostly comes from packaging applications (~62% in Europe) [56]. Amongst all plastic items, packaging items are some of the most complicated ones from a composition point of view. Most of them are multilayered materials and, therefore, are difficult to separate and purify. In food packaging, multilayer films composed of two or more polymers are often used to combine the advantages of the components, thus improving their overall performance. For example, multilayer films containing PET/poly(vinylidene dichloride)/PE with a thickness ratio of 25/5/70 are used in processed meat (e.g., ham) packaging [57]. For this reason, the greatest part of discarded packaging materials is usually landfilled. As already discussed in the previous sections, the municipal packaging waste from recycling bins is delivered to waste selection facilities, where it is sorted into various materials. Some of them can be recycled, but a good part is discarded. The portion of plastic material that is rejected depends on the market trends and on the efficiency of the separation plant, but, on a general level, this large portion is made up of a heterogeneous plastic mixture, mainly PE, PP, PS, PVC and PET, contaminated by foreign materials, such as aluminum, paper, adhesives and textile. The best-case scenario for these post-consumer waste materials streams involves pyrolysis processes, which allows the partial recovery of the energy embedded into the chemical bonds of these organic polymers. However, the largest part of these wastes goes to landfill. The main hurdles in the reuse of these mixtures come from the fact that the components are usually immiscible. The easiest and most economical way of recycling these mixtures would be to simply melt and reprocess them. However, the resulting melt reprocessed/recycled blends usually possess poor mechanical properties due to these immiscibility issues [58]. As previously stated, it is well known that polymers are usually not miscible, especially when they have significantly different chemical natures (for example, polyolefins and polyesters). This scarce compatibility ends up in phase separation phenomena because the interfacial adhesion between the different species is poor [59,60].

To achieve recycled blends between immiscible polymers with desirable mechanical properties, compatibilizer additives are typically required. These species are usually based on random, block or graft copolymers containing chemical units that either (1) interact strongly with or (2) resemble the primary polymer blend components. One of the key elements of these compatibilizers is that they have portions that are partially or completely miscible with each component of the blend. If designed appropriately, compatibilizer additives are able to localize at the interface between the different components of the blends resulting in a decrease in the interfacial energy and enhancement in interfacial adhesion between the immiscible blend polymers [61]. The result of this action exerted by the compatibilizer is that the minority phase is more compatible with the rest of the material, and it is, therefore, able to be finely dispersed within the matrix. This behavior ends up in the formation of a more homogeneous material that virtually behaves as the result of a combination of miscible components. As a result, the macroscopic mechanical properties of immiscible blends can be enhanced and expected to be in between those of the neat homopolymer blend components [62].

Alongside compatibilizer additives lie layer polymers that can also be developed. They are primarily designed as intermediary layers for improving interlayer adhesion in multilayer films. Their potential compatibilization capabilities in an immiscible blend are not taken into consideration for their applicability; however, they share chemical similarities with more classical compatibilizer additives.

Another alternative is represented by reactive compatibilization strategies. This approach involves the use of functionalized polymers based on the chemical nature of the main component of the mixture. The different functional groups are inserted along the chains in order to be able to react with the other components of the mixture during the various processing steps at high temperatures. One of the most significant examples is represented by the compatibilization of PET-PE mixtures. PE-based polymers are synthesized bearing different functionalities, such as maleic anhydride, glycidyl or amino functional groups, that are able to react with the PET chain ends during the extrusion [63,64].

## 5. A Case Study: The Italian Plasmix

According to the Italian plastic waste management stream, Plasmix is defined as what remains after the sorting of the most easily separable and recyclable plastics. It derives from the undersieve parts that are collected after the first phases of sorting and by the residues deriving from other mechanical operations performed during the recycling stream of waste plastic (Figure 3). This implies that the Plasmix composition changes according to many different factors and even according to the specific sorting plant. Albeit having a non-well-defined composition, Plasmix is mainly composed of PE and PP and most frequently contains small fractions of PS, PET and other polymers, as well as inerts. Therefore, some scientific research has been performed in order to find possible future uses. The Life Cycle Assessment (LCA) of Plasmix was investigated some years ago, considering different possible scenarios for its use, namely thermal treatment, energy recovery and landfilling [65], but no use in a circular economy framework has been considered. More recently, its reuse for bituminous pavements was assessed [66], as well as the aforementioned thermal treatment/pyrolysis [15], and it was studied as an alternative fuel in clinker production in order to reduce CO_2_ emissions [67].

Moreover, the interest in the possible production of syngas deriving from a mixture of Plasmix and refuse-derived fuels led to some studies in this direction [68]. All these studies are very recent, thus confirming the pressing need for finding a valid use of Plasmix as a secondary raw material, and most of them are focused on its use for the production of energy and syngas, showing that its potential as a raw material for mechanical recycling (i.e., secondary recycling) has not been exploited yet.

## 6. Environmental Concerns

The mass production of plastic began in the 1940s and since then has seen exponential growth, reaching the global annual total of 348 million tons (Mt) in 2017 [2]. It has been estimated that only the 30% of all plastics that have been produced are currently in use, while over 5800 Mt of plastic waste has been generated across the world [1]. This waste is managed in different ways, with landfilling, recycling and incineration being the most common (see above). However, the mismanagement of the waste coupled with the presence of illegal dumping sites can lead to the release and accumulation of plastics in environments [69]. Although the fate of plastic waste can vary both geographically and through time, its release into the environment is global and has occurred since the early period of mass production [70]. Nowadays, plastic waste has been found all over the world, not only in surface waters and sediments from marine and freshwater environments [71] but also in remote areas, such as deep sea [72] and glaciers [73,74]. Overall, the plastic waste found in the environment can be categorized according to its size in: macroplastics (items > 25 mm), mesoplastics (items 5–25 mm), large microplastics (items 1–5 mm), small microplastics (items 1 μm–1 mm) and nanoplastics (items 1–1000 nm) [75]. Among these size classes, different studies have highlighted the hazardousness posed by macro and mesoplastics towards different organisms [76], while the investigation of items with smaller sizes is more difficult and controversial [75,77]. In the environment, plastic waste can experience weathering processes that induce their fragmentation to smaller items (i.e., micro- and nanoplastics) following different pathways, such as photo-degradation, ultraviolet radiation or photo-oxidation [77]. Thus, while the total mass of plastic waste does not increase, the number of particles rises exponentially with decreasing size. These smaller items are not stationary but can move among different environmental compartments according to their physical-chemical (i.e., density) characteristics [78] and, thanks to their smaller size, they are bioavailable for ingestion by a wide range of organisms, both aquatic and terrestrial, as they overlap with the size range of their food [79]. Several studies evidenced the capability to ingest plastic waste (manly micro- and nano-plastics) by a wide range of organisms [80,81,82,83,84,85,86]. The ingestion could result in different effects such as the imbalance of the oxidative status [80,81,83,85], histological damage and inflammatory response [80,81], as well as changes in the growth rate and some behavioral traits [83,84,86]. Although the presence and potential effects of plastic waste were confirmed by a growing number of studies, these investigations are still in their infancy and deserve to be further improved. Overall, the threats posed by the release of plastics to the environment involve several different aspects and are yet to be fully understood and described. Indeed, to date, there is a wide degree of uncertainty concerning the environmental concentrations of plastics and the consequences of their presence toward organisms. Therefore, due to its complexity, the environmental issue of plastics could be seen as a wicked problem that needs to be further investigated in order to shed light on the impact of these contaminants on the ecosystems.

## 7. Conclusions

The present review summarizes the current strategies related to the management of plastic waste, pointing out the challenge of the non-homogeneous, non-recyclable plastic waste. Some of the most significant sorting strategies for the separation of the different kinds of plastic materials in the waste stream have been highlighted and described. The importance of the sorting process in plastic waste management has been underlined, as it profoundly affects the efficiency of the recycling methodologies. Primary, secondary, tertiary and quaternary recycling approaches have been described, pointing out the advantages and shortcomings. As incineration and landfills are two unsustainable practices for the management of plastic waste because they lead to remarkable secondary pollution of ecosystems and loss of economic value for materials that could be reused or recycled profitably, the improvement of recycling methods represents a priority in plastic waste management. Despite the recent progress, the greatest part of plastic waste cannot be recycled, ending up in the generation of heterogeneous mixtures of materials, the only fate of which nowadays is incineration. The development of an efficient recycling strategy for these complex mixtures of wastes would be particularly relevant from both an industrial and environmental point of view. The case study of the Plasmix in the Italian scenario has been reported, pointing out the need to develop alternative and profitable approaches for the valorization of these wastes, preventing incineration, landfill disposal or environmental contamination. To date, a limited number of studies looked for valuable use of the mixed fraction of plastic waste in different industrial fields, but none investigated the opportunity to manage these raw materials for mechanical recycling (i.e., secondary recycling). Considering the huge amount of non-recyclable plastic waste conferred to the sorting plants, new and valuable approaches and/or strategies of reuse or recycling needs to be developed and optimized in order to profitably and sustainably manage these wastes. Finally, the concerns related to the plastic waste accumulation in the environment have been discussed with particular focus on the micro- and nanometric-sized items that are generated from plastic debris after prolonged exposure to weathering. However, the adverse effects induced by the exposure to micro- and nanoplastics on free-living organisms still have to be fully understood and evaluated.

## Figures and Tables

**Figure 1 ijerph-19-10088-f001:**
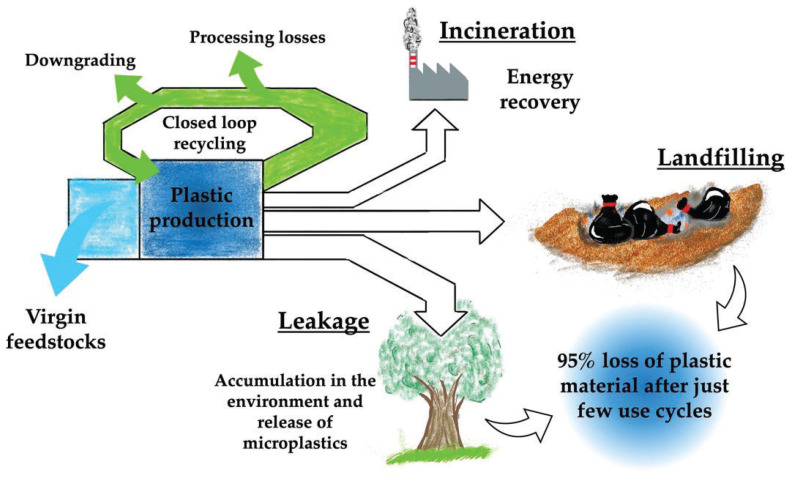
Schematic representation of the life cycle of plastic materials.

**Figure 2 ijerph-19-10088-f002:**
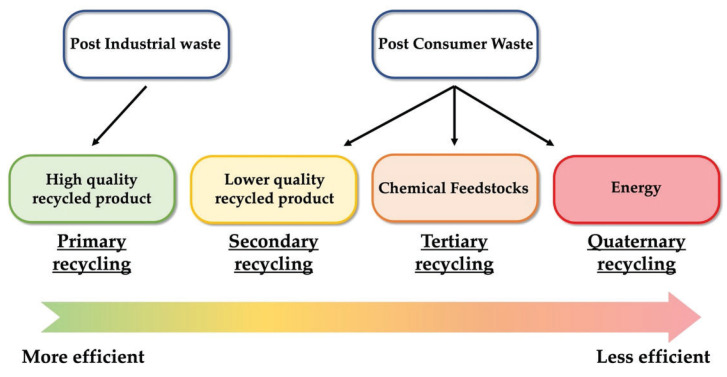
Common classification of usual recycling strategies for plastic wastes.

**Figure 3 ijerph-19-10088-f003:**
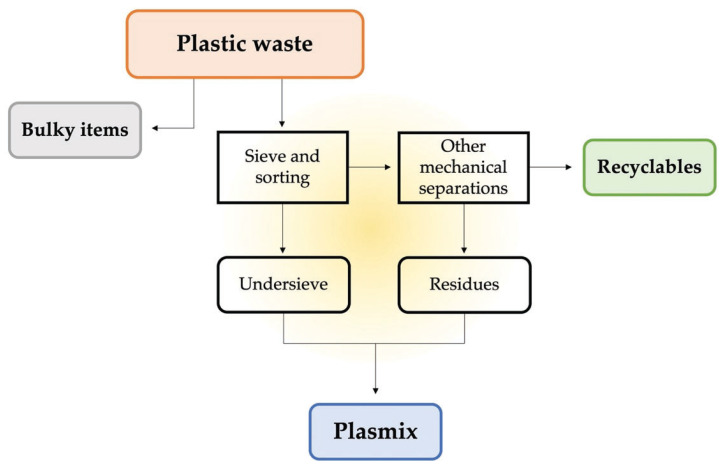
Pathways of Plasmix generation from plastic waste in Italy.

## Data Availability

Not applicable.

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
