# Peer review of "Approaches for Management and Valorization of Non-Homogeneous, Non-Recyclable Plastic Waste"

_ijerph, 2022, doi:10.3390/ijerph191610088_

Round 1
Reviewer 1 Report
Report on the manuscript
Approaches for management and valorization of non-homogeneous, non-recyclable plastic waste
Stefano Gazzotti , Beatrice De Felice , Marco Aldo Ortenzi , Marco Parolini
Manuscript ID: ijerph-1842188
The presented study aims to create an overview of environmental issues. Plastic waste management is defective globally.
The paper presents in detail the difficulties of managing non-recyclable materials, such as the mechanical or chemical methods used, the recycling or reusing of textile materials for secondary recycling, the Italian method of obtaining Plasmix plastic products, as well as the limitations encountered. This topic is current, as concerns about the environment are presented, being analyzed the long-term impact of the integration of plastic masses in the environment, through the appearance of microplastic particles.
Before that the Editor makes a decision I suggest that the authors take into account the following corrections:
1. The author could expand the conclusions part.
2. Please check the text editing. The figure caption should start with capital letter, and there is a highlighted reference number in the references section.
3. I suggest adding some visual descriptions (images, graphs) of some of the processes, methods and statistical data described in the paper.
If the author takes into account these observations the work can be published.
Author Response
REPLY to reviewer #1
Approaches for management and valorization of non-homogeneous, non-recyclable plastic waste
Stefano Gazzotti , Beatrice De Felice , Marco Aldo Ortenzi , Marco Parolini
Manuscript ID: ijerph-1842188
The presented study aims to create an overview of environmental issues. Plastic waste management is defective globally.
The paper presents in detail the difficulties of managing non-recyclable materials, such as the mechanical or chemical methods used, the recycling or reusing of textile materials for secondary recycling, the Italian method of obtaining Plasmix plastic products, as well as the limitations encountered. This topic is current, as concerns about the environment are presented, being analyzed the long-term impact of the integration of plastic masses in the environment, through the appearance of microplastic particles. Before that the Editor makes a decision I suggest that the authors take into account the following corrections:
- The author could expand the conclusions part.
REPLY: thank you for your suggestions, we enlarged the conclusion section as follows: “The present review summarizes the current strategies related to the management of plastic waste, pointing out the challenge of the non-homogeneous, non-recyclable plastic waste. Some of the most significant sorting strategies for the separation of the different kinds of plastic materials in the waste stream have been highlighted and described. The importance of the sorting process in the plastic waste management has been underlined, as it profoundly affects the efficiency of the recycling methodologies. Primary, secondary, tertiary and quaternary recycling approaches have been described, pointing out the adavantages and the shortcomings. As incineration and landfills are two unsustainable practices for the management of plastic waste because they lead to remarkable secondary pollution of ecosystems and loss of economic value for materials that could be reused or recycled profitably, the improvement of recycling methods represents a priority in plastic waste management. Despite the recent progresses, the greatest part of plastic waste cannot be recycled, ending up in the generation of heterogeneous mixtures of materials which only fate nowadays is incineration. The development of an efficient recycling strategy of these complex mixtures of wastes would be particularly significant from both an industrial and environmental point of view. The case study of the Plasmix in the Italian scenario, have been reported, pointing out the need to develop alternative and profitable approaches for the valorization of these wastes, preventing the incineration, landfill disposal or environmental contamination. To date, a limited number of studies looked for a valuable use of mixed fraction of plastic waste in different industrial fields, but none investigated the opportunity to manage these raw materials for mechanical recycling (i.e., secondary recycling). Considering the huge amount of non-recyclable plastic waste conferred to the sorting plants, new and valuable approaches and/or strategies of reuse or recycling need to be developed and optimized in order to profitably and sustainably manage these wastes. Finally, the concerns related to the plastic wastes accumulation in the environment have been discussed with particular focus on the micro and nanometric-sized fragments that generate from plastic debris after prolonged exposure to weathering. To this regard, the investigation of the effects of said fragments on living being still has to be fully understood and evaluated.”
- Please check the text editing. The figure caption should start with capital letter, and there is a highlighted reference number in the references section.
REPLY: Thank you for your suggestion. Figure captions now start with a capital letter and we removed the highlighted number in reference list.
- I suggest adding some visual descriptions (images, graphs) of some of the processes, methods and statistical data described in the paper.
REPLY: we included and additional Figure (Figure 1) showing a schematic representation of the life cycle of plastic materials. No figures on methods and statistical analyses can be included because no statistical analyses were performed.
If the author takes into account these observations the work can be published.
REPLY: thank you for your positive comments that helped us to improve the manuscript.
Reviewer 2 Report
attached in pdf

Author Response
REPLY to reviewer #2
Review report
Manuscript ID: ijerph-1842188
Type of manuscript: Review
Title: Approaches for management and valorization of non-homogeneous, non-recyclable plastic
waste
This presented review is to return a comprehensive overview on environmental and management issues related to the complex and heterogeneous mixture of plastic wastes that is generated at the end of the sorting procedures in Italian plastic recycling plants, the so- This study very well described new strategies to tackle accumulation of plastic waste is of crucial importance and different approaches are being investigated to effectively reduce plastic wastes generated by an improper or inefficient disposal. The manuscript is very well written and constructed including storytelling. Chapters are clearly formatted and descripted with interesting facts. Therefore, this manuscript cannot be considered for acceptance without a minor revision.
REPLY: thank you for your positive comments. We are please you consider our manuscript as interesting, very well written and structured.
- Why authors describe microplastic of range of the 1 to < 1,000 µm size? The NOAA (National Oceanic and Atmospheric Administration) proposed a working definition in which microplastics are all plastic particles <5 mm in diameter, which has become the most frequently used definition (L58).
REPLY: we preferred to use the recent definition of microplastics suggested by Hartmann and coauthors because it is more focused and detailed.
Hartmann, N. B.; Huffer, T.; Thompson, R. C.; Hassellöv, M.; Verschoor, A.; Daugaard, A. E.; Wagner, M. Are we speak-ing the same language? Recommendations for a definition and categorization framework for plastic debris. Environ. Sci. Technol., 2019, 53, 1039–1047. DOI: https://doi.org/10.1021/acs.est.8b05297
- Additional figures will enrich the manuscript and improve the entered data (L64-70)
REPLY: as you suggested we included and additional Figure (Figure 1) showing a schematic representation of the life cycle of plastic materials.
- Authors should add information about the uniqueness and research gap that fill the above research
REPLY: Thank you for your suggestion. We included at the end of the introduction section a brief paragraph summarizing the aims of and the novelty of our reviews as follows: “This review lists the different techniques currently available for sorting and recycling of plastic waste and first focuses on the difficulties and limitations related to the manage-ment of non-homogeneous, non-recyclable plastics (i.e., Plasmix), highlighting the strate-gies for the proper, sustainable and valuable use of this plastic waste”.
- Authors should increase sub-sections (2.1-2.5) about advantages, disadvantages and challenges of
each sorting methods
REPLY: Thank you for your suggestion. At the end of each sub-section concerning the recycling techniques we included advantages and limitations for each specific technique.
- That chemical recycling its common or sill development method? What about of economic and environmental aspects of chemical recycling? (Section 3.3)
REPLY: thank you for your suggestion. At the end of the additional paragraph concerning advantages and limitations of chemical recycling we included a sentence on economic/environmental aspects as follows: “To this regard, also tertiary recycling is affected by the purity of the starting material and it could, in principle, turn the process into a non-convenient approach from an economical point of view. In addition, chemical recycling usually requires solvents and chemicals that could represent a threat for the environment”.